# Mobile Charging Sequence Scheduling for Optimal Sensing Coverage in Wireless Rechargeable Sensor Networks

Jinglin Li [1,2], Chengpeng Jiang [1,2], Jing Wang [3], Taian Xu [4,*] and Wendong Xiao [1,2,3,*]

1 School of Automation and Electrical Engineering, University of Science and Technology Beijing, Beijing 100083, China
2 Beijing Engineering Research Center of Industrial Spectrum Imaging, Beijing 100083, China
3 Shunde Innovation School, University of Science and Technology Beijing, Shunde 528399, China
4 Zaozhuang University, Zaozhuang 277160, China
* Correspondence: 101091@uzz.edu.cn (T.X.); wdxiao@ustb.edu.cn (W.X.)

**Abstract:** In wireless rechargeable sensor networks (WRSNs), a novel approach to energy replenishment is offered by the utilization of mobile chargers (MCs), which charge nodes via wireless energy transfer technology. However, previous research on mobile charging schemes has commonly prioritized charging efficiency as a performance index, neglecting the importance of quality of sensing coverage (QSC). As the network scale increases, the MC's charging power becomes unable to meet the energy needs of all nodes, leading to a decline in network QSC when nodes' energy is depleted. To solve this problem, we study the problem of mobile charging sequence scheduling for optimal network QSC (MSSQ) and propose an improved quantum-behaved particle swarm optimization (IQPSO) algorithm. With the attraction of potential energy in quantum space, this algorithm will adaptively adjust the contraction expansion coefficient iteratively, leading to a global optimal solution for the mobile charging sequence. Extensive simulation results demonstrate the superiority of IQPSO over the widely used QPSO and Greedy algorithms in terms of network QSC, especially in large-scale networks.

**Keywords:** wireless rechargeable sensor networks; quality of sensing coverage; mobile charging sequence scheduling; contraction expansion coefficient; improved quantum-behaved particle swarm optimization

## 1. Introduction

Wireless sensor networks (WSNs) are widely applied in various fields due to their low cost, scalability, self-organizing dynamics, and fault tolerance characteristics [1], such as environmental monitoring [2], medical treatment [3], elderly care services [4], intelligent transportation [5], and manufacturing systems [6]. Despite these benefits, the development of WSNs is hindered by the limited battery capacity of the sensors [7]. Furthermore, frequent battery replacement is often cumbersome, costly, dangerous, and unrealistic.

To address the problems mentioned above, researchers have made nodes passively collect solar [8], wind [9], or thermoelectric [10] energy from the environment in energy harvesting WSNs to prolong the network lifetime. However, these approaches could be more stable and predictable. In recent years, with the advancement of wireless power transfer technology, a new concept of wireless rechargeable sensor networks (WRSNs) has emerged, where mobile chargers (MC) with a large capacity provide a highly reliable and efficient energy supplement to the nodes. Therefore, the study of MC charging strategies in WRSNs has recently gained significant attention [11–15].

In practical network applications, the quality of information (QoI) is essential in network performance. Different performance indexes are used to evaluate network QoI for different applications, including quality of coverage [16,17] in environmental monitoring networks, tracking accuracy [8] in target tracking networks, and detection probability [18,19]

in event detection networks. The energy consumption of nodes in the network is different. If the MC charging sequence scheduling is unreasonable, the nodes will run out of energy because they cannot be charged in time. It leads to the vulnerabilities in network sensing coverage, information loss, and reduced reliability. Although the optimization of QoI in WSNs has become more sophisticated [20], more research needs to be conducted on WRSNs. As a critical factor in evaluating the network coverage performance, quality of sensing coverage (QSC) is the foundation of network credibility. This paper studies the novel problem of mobile charging sequence scheduling for network QSC (MSSQ); how to optimize the mobile charging sequence of MC to maximize the network QSC.

In the MSSQ problem, the state of all nodes changes correspondingly based on the same MC charging decisions made at different times, unlike in the traveling salesman problem. Thus, the MSSQ problem can be formulated as an extended traveling salesman problem (ETSP). Quantum-behaved particle swarm optimization (QPSO) is a novel optimization algorithm based on swarm intelligence with the potential to find practical solutions to NP complete problems due to the gravitational effect of quantum potential energy. To enhance the convergence efficiency of the algorithm, we present an improved QPSO (IQPSO) approach, which adaptively adjusts the contraction expansion coefficient during iterations to optimize convergence speed while ensuring convergence.

The contributions of the paper are summarized as follows:

(1) In this paper, the impact of nodes' sensing coverage contributions at different times on network QSC is considered in WRSNs, and a novel mobile charging sequence scheduling for network QSC (MSSQ) problem is studied.

(2) The MSSQ problem is formulated as an ETSP based on a new performance index for network QSC, named the total sensing coverage and node survival rate ($TSCNS$), which considers the sensing coverage performance and node survival rate.

(3) An effective IQPSO algorithm is proposed for MSSQ to obtain the suboptimal mobile charging sequence faster, to avoid falling into local optima.

The rest of the paper is organized as follows: Section 2 introduces the related work. The mobile charging WRSNs and sensing coverage models are described in Section 3. The MSSQ problem is formulated mathematically in Section 4. The IQPSO algorithm for MSSQ is presented in Section 5, along with its convergence property. Simulation results are reported in Section 6 to demonstrate the performance of the IQPSO. Discussions are presented in Section 7. Conclusions and the future work are given in Section 8.

## 2. Related Work

Wireless charging of WRSNs can be realized in two methods: fixed-point charging and mobile charging. In the former, the fixed charger is placed in advance to replenish the node's energy [21]. Charging distance restricts the energy conversion efficiency compared with the fixed-point charging method; therefore, related research on the mobile charging scheduling problem has also made significant progress recently due to its flexible and efficient charging characteristics in WRSNs, including offline and online methods.

### 2.1. Mobile Charging Methods

The offline method assigns the determined charging sequence to MC according to the specified path when the network state is known. Wei et al. [14] constructed a multi-objective optimization MC charging path planning model, under which MC can supplement energy and collect data simultaneously. They proposed a multi-objective ant colony optimization algorithm to minimize the average data transmission delay. In order to optimize the charging timeliness of nodes, Jiang et al. [12] defined a secondary performance index of charging waiting time and used it to evaluate MC charging performance. In order to maximize the mobile charging efficiency, Mo et al. [13] transformed the multi-MC coordination problem into a mixed integer linear programming problem to minimize the MC energy consumption and ensure that each node will not run out of energy. In the multi-node charging scheme, MC moves to the clustered charging location

and charges all nodes within the charging range. Although the charging efficiency of MC for a single node is reduced under this scheme, the overall charging efficiency is significantly improved [22]. Han et al. [23] presented an uneven cluster-based mobile charging algorithm, which determines the nodes to be charged through the uneven network clusters to reduce the number of off-working nodes. Sun et al. [24] took the critical path as the core factor and proposed a decomposition strategy to identify the interacting variables based on the CP. Finally, they combined the grouping strategy and cooperative evolutionary algorithm and proposed a new algorithm, to optimize the power allocation strategy in the system. Using multiple MCs for multi-node charging, Xu et al. [25] determined each MC's independent closed charging path by designing the problem of minimizing the maximum charging delay to accelerate the charging speed and reduce the number of dead nodes.

The offline method assumes that the energy consumption of networks changes periodically. However, this method is challenging to implement in practical applications because the network state is dynamic. In the online method, when the residual energy of the node is lower than a certain threshold, it will immediately send a charging request to the MC. MC adjusts the charging action according to the real-time state of WRSNs. He et al. [26] used a simple and effective nearest-job-next with preemption algorithm to determine the MC's charging decision according to the nodes' position in the request queue. In the same year, Lin et al. [27] proposed a double warning threshold with double preemption algorithm, which uses two warning thresholds to adjust the charging priority of different nodes. In addition, Lin et al. [28] presented a primary and passer-by scheduling algorithm, which considers the urgency and proximity of balanced charging and ignores the effect of selecting inefficient nodes. Considering the influence of network topology changes, node failures and other uncertain factors on the charging performance, Lin et al. [14] proposed a temporal spatial charging scheduling algorithm for large-scale WRSNs which calculated the node's dead time according to the node's energy state and adjusts the charging sequence according to the dead time.

### 2.2. Network QoI Optimization

Most strategies reduce network energy consumption by scheduling nodes to ensure network QoI under limited network energy [20]; research on QoI optimization in WSNs is limited. Moreover, Gaudette et al. [7] optimized and controlled each node's sensing range to maximize the network QSC. Liu et al. [8] proposed a new multi-step prediction-based adaptive dynamic programming method for cooperative target tracking. For the research of target coverage, Xiong et al. [18] presented a two-stage lifetime enhancement method. In the directional rechargeable network, Zhu et al. [16] redeployed the network distribution when the sensing angle and energy of the directional node were limited. Xu et al. [19] proposed a maximizing cooperative detection probability algorithm for barrier coverage to maximize the detection probability of the network.

In summary, there is a gap in the existing research on the optimization of mobile charging sequences for network sensing coverage performance. In this paper, from the perspective of the known state of WRSNs, we propose an IQPSO algorithm to find the optimal mobile charging sequence ensuring the maximum network QSC .

### 3. System Models

Figure 1 gives an example of the schematic diagram of the mobile charging WRSN model, including one sink node (SN), charging station (CS), MC and a set of $N$ rechargeable randomly deployed nodes $S = \{s_1, s_2, \ldots, s_N\}$. The positions of the nodes $(x_{s_i}, y_{s_i})$ are fixed and known $1 \leq i \leq N$. CS is responsible for replenishing energy for MC. SN collects the real-time information of all nodes and formulates the mobile charging sequence. According to the established charging scheduling sequence, MC starts from the CS, moves near to the node for charging one by one according to the schduling decision, then returns to the depar-

ture when all nodes have been charged. We assume that the network deployment scenarios are barrier-free, and accessible. Symbols in this paper are summarized in Nomenclature.

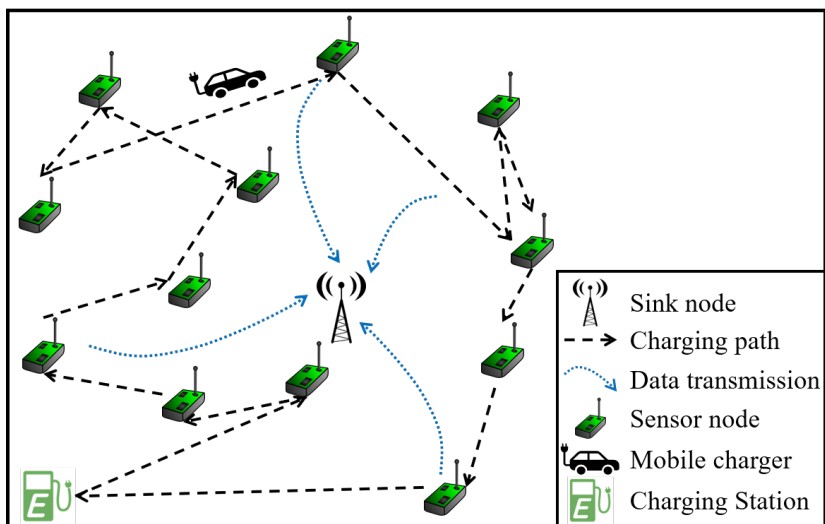

**Figure 1.** Schematic diagram of the mobile charging WRSN model.

**Definition 1.** *Charging time step, CTS: the time step when MC is ready to charge or has finishing charging.*

The following describes the model assumptions and conventions from three aspects: nodes, MC, and network sensing coverage.

**Assumption 1.** *Sensor Node: Each node has the same function and can predict their own power consumption and residual energy according to their data acquisition frequency and data transmission rate [12], which is expressed as $V_{cs} = \{v_{cs}^1, v_{cs}^2, \ldots, v_{cs}^N\}$ and $E_r = \{e_r^1, e_r^2, \ldots, e_r^N\}$. In this paper, nodes will stop working if they run out of energy and resume working after being charged.*

At the m CTS, the real-time residual energy of the WRSNs is expressed as:

$$E_r(m) = E_r(m-1) - V_{cs}t_w(m-1),  \tag{1}$$

where $t_w(m)$ is the total working time of MC, including the MC moving time, $t_l(m-1)$, and charging time, $t_c(m-1)$, at the (m-1) CTS, which is represented by $t_w(m) = t_l(m-1) + t_c(m-1)$. Assume that $\Phi$ is the MC sequence. At the m CTS, $s_{\Phi(m)}$ is selected to charge, and we know that $t_l(m) = d(m)/v_m$ and $t_c(m) = e_m - e_r^{\Phi(m)}(m)/v_c$. We can see that $d(m)$ is the Euclidean distance between $s_{\Phi(m-1)}$ and $s_{\Phi(m)}$, and $D(m)$ is the set of distances between all nodes.

*3.1. Mobile Charging Model*

**Definition 2.** *Charging cycle: the process of MC leaving the CS to perform the charging task and then returning is called a charging cycle.*

**Assumption 2.** *MCs: MCs can only charge one node at the same time and leave when the node is fully charged. MC charging power, moving speed, and energy consumption per unit moving distance are all fixed values. The maximum $E_m$ satisfies the required energy to complete the mobile charging sequence.*

At the m CTS, the MC chooses $s_j$ to charge, so the real-time MC residual energy, $e_{MC}(m)$, is expressed as

$$e_{MC}(m) = e_{MC}(m-1) - (e_m - e_r^j(m-1)) - v_l t_l(m-1).  \tag{2}$$

### 3.2. Sensing Coverage Model

**Assumption 3.** *Node deployment*: *incomplete or redundant sensing coverage of the network will reduce network QSC. In this paper, to avoid network over-sensing coverage redundancy, the area of cooperation sensing by three covered nodes will not be covered by the fourth node again.*

The distribution of network nodes leads to complex intersecting coverage of multiple nodes. Under different CTS, the entire sensing coverage area of the WRSN changes due to node energy exhaustion and replenishment. This paper provides an accurate real-time sensing coverage area calculation scheme. A few variables are introduced first to facilitate the introduction of this scheme. $\triangle ij$ is the triangle formed by the intersection of $s_i$ or $s_j$ and two nodes. $\triangle ijk$ is the triangle formed by the intersection of $s_i$, $s_j$, and $s_k$ within the three intersecting sensing areas. $h^i_{\triangle ijk}$ is the edge corresponding to $s_i$ in $\triangle ijk$. $a^i_{\triangle ijk}$ is the triangle area enclosed by $s_i$ and $h^i_{\triangle ijk}$. $\overset{\frown}{h}{}^i_{\triangle ijk}$ is the circular arc of $h^i_{\triangle ijk}$. $a_{\triangle ij}$ and $a_{\triangle ijk}$ are the areas of $\triangle ij$ and $\triangle ijk$, respectively. $a^i_{\triangle ij}$ is the triangular area enclosed by $s_i$ and $h^i_{\triangle ijk}$. $a_{\overset{\frown}{h}{}^i_{\triangle ijk}}$ is the sector area enclosed by $s_i$ and $\overset{\frown}{h}{}^i_{\triangle ijk}$. $\angle S_{ij}$ is the $s_i$ top angle in $\triangle ij$ and $\angle S_{ijk}$ is the $s_i$ top angle in $\triangle ijk$.

**Definition 3.** *Area of Cooperative Sensing, AoCS*: *the sensing coverage area covered by multiple nodes in the WRSN is defined as the AoCS, and the set is expressed as $A_c = \left\{ a^1_c, a^2_c, \ldots, a^N_c \right\}$. The specific calculation method of the AoCS is shown in Algorithm 1.*

**Definition 4.** *Area of Independent Sensing, AoIS*: *the area independently covered by the node after removing the $a_c$ is called the AoIS, and the set is $A_I = \left\{ a^1_I, a^2_I, \ldots, a^N_I \right\}$.*

*The total AoIS of the network is expressed as $a_{TI} = \sum\limits_{j=1}^{N} a^j_I = \sum\limits_{j=1}^{N} \left( a_m - a^j_c \right)$, $a_m = \pi (R_m)^2$.*

*Under the m CTS, nodes may stop working if they run out of energy, so the WRSN real-time total sensing area (TSA) changes, denoted as $a_{TSR}(m)$. The specific calculation is shown in (3)–(6), where $a^i_v$ represents the $s_i$ average AoCS.*

$$a^i_v(m) = \left\{ \sum_{k_2=1}^{l_{Q_i}} a^{i,Q_i(k_2)}_c(m) - \sum_{k_1=1}^{l_{q_i}} a^{i,q_i(k_1)}_c(m) \right\} / 2 - \left\{ \sum_{k_1=1}^{l_{q_i}} a^{i,q_i(k_1)}_c(m) \right\} / 3 \tag{3}$$

$$a_{TC}(m) = \sum_{i=1}^{N} a^i_v(m). \tag{4}$$

$$a^i_I(m) = a_m - \left\{ \sum_{k_2=1}^{l_{Q_i}} a^{i,Q_i(k_2)}_c(m) - \sum_{k_1=1}^{l_{q_i}} a^{i,q_i(k_1)}_c(m) \right\}. \tag{5}$$

$$a_{TSA}(m) = a_{TC}(m) + a_{TI}(m) = \sum_{i=1}^{N} \left\{ a^i_v(m) + a_m - \left\{ \sum_{k_2=1}^{l_{Q_i}} a^{i,Q_i(k_2)}_c(m) - \sum_{k_1=1}^{l_{q_i}} a^{i,q_i(k_1)}_c(m) \right\} \right\} \tag{6}$$

---

**Algorithm 1** Real-time AoCS Calculation.

---

1: **INPUT:** $E_r(m)$, $D(m)$, $R_m$

2: **Outputs:** $A_c(m)$

3: For $i = 1 : (N - l_s(m))$

4:     Find $Q_i$, $q_i$, $l_{Q_i}$ and $l_{q_i}$

5:     If $Q_i = \varnothing$

6:         $a_c^i(m) = 0$

7:     Else

8:         For $j = 1 : l_{Q_i}$

9:             Calculate the $\angle S_{j,i}$ of $s_j$

10:             $a_{\triangle ij} = R_m{}^2 sin(\angle S_{ji}) \cos(\angle S_{ji})$;

11:             $a_c^{i,j} = \pi R_m{}^2 ((\angle S_{j,i})/360°) - a_{\triangle ij}$

12:             If $s_j$ is not in $q_i$

13:                 $a_c^{i,j,k} = 0$;

14:             Else

15:                 Find another three-intersecting node $s_k$

16:                 If double-count

17:                     $a_c^{i,j,k} = 0$

18:                 Else

19:                     Center angle $\angle S_{j,i,k}$ of $a_c^{i,j,k}$ in $s_j$ can also be calculated

20:                     So as $\angle S_{i,j,k}$ and $\angle S_{k,i,j}$; $h_{\triangle ijk}^i$, $h_{\triangle ijk}^j$ and $h_{\triangle ijk}^k$ can also be obtained

21:                     $ss = (h_{\triangle ijk}^i + h_{\triangle ijk}^j + h_{\triangle ijk}^k)/2$

22:                     $a_{\triangle ijk} = \sqrt{ss(ss - h_{\triangle ijk}^i)(ss - h_{\triangle ijk}^j)(ss - h_{\triangle ijk}^k)}$

23:                     Take $h_{\triangle ijk}^i$ as an example

24:                     $h_{\triangle ijk}^i = \sqrt{(2R_m{}^2 - \cos \angle S_{ijk})/2R_m{}^2}$

25:                     The three-intersection triangle and sector area of $s_i$

26:                     $a_{\triangle ijk}^i = h_{\triangle ijk}^i (\sin(\angle S_{ijk}/2)R_m)/2$; $a_{\overset{\frown}{h}{}_{\triangle ijk}^i} = \pi(R_m)^2(\angle S_{ijk})/360°$

27:                     $a_{\triangle ijk}^j$, $a_{\triangle ijk}^k$, $a_{\overset{\frown}{h}{}_{\triangle ijk}^j}$ and $a_{\overset{\frown}{h}{}_{\triangle ijk}^k}$ can also be solved

28:                     $a_c^{i,j,k} = a_{\triangle ijk} + a_{\overset{\frown}{h}{}_{\triangle ijk}^i} + a_{\overset{\frown}{h}{}_{\triangle ijk}^j} + a_{\overset{\frown}{h}{}_{\triangle ijk}^k} - a_{\triangle ijk}^k - a_{\triangle ijk}^i - a_{\triangle ijk}^j$

29:                 End

30:             End

31:         End

32:     End

33:     $a_c^i = \sum\limits_{j=1}^{l_{Q_i}} (a_c^{i,j} - a_c^{i,j,k})$

34: End

---

## 4. Problem of Mobile Charging Sequence Scheduling for Network QSC

In this section, we describe the MSSQ problem and prove that MSSQ is NP-complete.

### 4.1. Problem Formulation

With the increasing of the number of nodes, more nodes may stop working if they run out of energy, because the charging capacity of MC cannot meet the network's total energy consumption. At this time, the network topology changes, and the coverage vulnerability makes the sensing information incomplete, thus reducing the network QSC. How to guarantee the maximum network QSC in unit charging cycle under the condition of the limited charging capacity of MC is the goal of the MSSQ problem. We redefine a new performance index for network QSC as sensing coverage and node survival rate (SCNS), which is explicitly expressed as $SCNS(\Phi(m)) = \delta_1 \frac{a_{TSA}(m)}{a_{tm}} + \delta_2 \frac{N - l_s(m)}{N}$,

where $\delta_1 = 0.8$ and $\delta_2 = 0.2$ are two performance parameters, $a_{tm}$ is the total area of sensing coverage of the network. We abstract the MSSQ problem as a multi-objective optimization problem with nonlinear discrete variables. The primary goal is to maximize the network sensing range, and the second is to improve the node survival rate. Therefore, maximizing network QSC is to maximize the sum of *SCNS* in unit charging cycle. The MSSQ problem is formulated as

$$
\begin{aligned}
Maximize \quad & TSCNS(\Phi) = \sum_{m=1}^{N} SCNS(\Phi(m)) \\
s.t. \quad & 0 \le e_r^i \le e_m \\
& 0 \le t_c(m)v_c \le e_m
\end{aligned}
. \tag{7}
$$

There are two constraints in MSSQ problem: the residual and supplemented energy of the node should be greater than 0 and not exceed the node capacity. The largest total *SCNS* (*TSCNS*) we get, the optimal mobile charging sequence $\Phi^*$ we find.

### 4.2. Proof of NP-Completeness

MSSQ is a multi-objective optimization problem with nonlinear discrete variables and an ETSP. TSP describes how a traveler starts from a specific city and designs a travel route to minimize the total travel distance back to the departure city after reaching all cities. This paper analyzes the particular case of MSSQ (MSSQ-P) through fixed conditions. We prove that the decision version of MSSQ-P is NP-complete, and MSSQ can also be proved.

**Theorem 1.** *The decision version in MSSQ-P is NP-complete.*

**Proof.** By simplifying the MSSQ, we prove that the decision version of MSSQ-P is NP-complete. Given a complete set of nodes $G_b = (E_b, V_b)$, $V_b$ contains a CS and N nodes to be charged, $E_b$ is weighted and expressed as the distance traveled between two nodes. The decision version of the TSP needs to find the shortest Hamiltonian loop $C'$ to cover all nodes in $V_b$ and to minimize the total weight of all links. Based on the above decision version, we construct instance $G_{b-p} = (E_{b-p}, V_{b-p})$ of MSSQ-P. $V_{b-p}$ contains a CS and a set of $N$ nodes. The link weight between two nodes in $E_{b-p}$ is less than or equal to 1, with dimension of N, representing the real-time *SCNS* generated by MC charging action under different CTS. In MSSQ-P, MC's capacity meets the network's overall energy demand, and the charging power reaches the maximum value of $v_{c-max}$ (nodes can be fully charged instantly). MC needs to charge all nodes in one charging cycle. Assuming that the energy consumption of nodes remains unchanged and that the node charged has sufficient endurance (the working time of the node is much longer than that of the charging cycle), the node in MSSQ-P only needs to be charged once. The purpose of MSSQ-P is to determine that there is a mobile charging sequence $\Phi_p^*$ to maximize the *TSCNS*. If the *TSCNS* of MC is N, $\Phi_p^*$ can get the optimal network QSC, the network is fully covered and there is no off-working nodes. Therefore, the optimal solution of MSSQ-P is the solution of ETSP, so we can prove that MSSQ-P is NP-complete.  □

Next, we need to prove the existence of the MSSQ-P solution. However, in WRSNs applications, the computation complexity of building the optimal mobile charging sequence for $N$ nodes is $O(N^2)$. When the state of the network node changes, the generated optimal mobile charging sequence $\Phi_p^*$ may no longer be optimal. For example, assuming that the optimal mobile charging sequence is $\Phi_p^* = \{s_1, s_2, \ldots, s_N\}$, we can construct another $\Phi'_p = \{s_N, s_2, \ldots, s_1\}$. When $e_r^N << e_r^1$ and $v_{cs}^N >> v_{cs}^1$, $s_N$ is likely to stop working in $\Phi_p^*$, the network QSC decreases. At this time, the *TSCNS* of $\Phi'_p$ is greater than $\Phi_p^*$.

## 5. The IQPSO Algorithm for MSSQ

Particle swarm optimization (PSO), a traditional heuristic algorithm, is usually used to solve NP-complete problems. Because the velocity of particles in the PSO

algorithm is always limited, particles cannot reach any point in the whole feasible space during the search process. Therefore, the PSO algorithm cannot converge to the optimal global solution, which is the defect of the PSO algorithm. To avoid the above situation, researchers put particles into quantum space and proposed the QPSO algorithm. At this time, the particle motion is described by the attraction generated by the quantum potential energy. Particles under quantum attraction can appear at any point in space with a certain probability. The optimal global solution can be obtained with a certain probability. Therefore, the QPSO algorithm has successfully solved various optimization problems [29–31].

In the QPSO algorithm, $\alpha$ is a parameter named 'contract expansion' coefficient [31] in updating the average optimal position of particles through quantum potential energy, which is concerned with the convergence of the QPSO algorithm. Therefore, for the MSSQ problem, we propose an IQPSO algorithm. In IQPSO, as the number of iteration steps increases, we adaptively adjust $\alpha$ according to the iterations to improve the performance of the QPSO algorithm.

### 5.1. Proposed IQPSO Algorithm

In the $N$-dimensional target search space, the IQPSO algorithm $\tilde{X} = \{X_1, X_2, \ldots, X_o\}$ consists of $o$ particle swarms, which represents the possible suboptimal mobile charging sequence of MSSQ problem. At the $k$ step, the position of the $i$-th particle is $X_i(k) = \{x_{i,1}(k), x_{i,2}(k), \ldots, x_{i,N}(k)\}$. In IQPSO algorithm, the particle does not have velocity vector, and the best position of the personal particle $i$ is expressed as $p_i = \{X_1, X_2, \ldots, X_o\}$. At the $k$ step, the $p_i(k)$ of the $i$-the particle is

$$p_i(k) = \begin{cases} X_i(k) & \Delta TSCNS \geq 0 \\ p_i(k-1) & \Delta TSCNS < 0 \end{cases}' \tag{8}$$

where $\Delta TSCNS = TSCNS(X_i(k)) - TSCNS(p_i(k-1))$. The position with the largest $TSCNS$ searched in particle represents personal experience represented by $p_{best}$, and set $p_{best} = p_i$ changing with particle optimization process. At the $k$ step, the global best position of the population $G_b$ is

$$m_1 = \arg \max_{1 \leq i \leq o} \{TSCNS(p_i(k))\}$$
$$G_b(k) = p_{m_1}(k) \tag{9}$$

where $m_1 \in \{1, 2, \ldots, o\}$ is subscript of the global optimal particle position in the population. The global best position should be calculated before each update of the particle's position. Thus, only the $TSCNS$ of the current personal position of each particle needs to be compared with the global best position, if the $p_i$ is better, update the $G_b$. To get the evolution equation of particles, the following settings are needed

$$p_{i,j}(k) = \varphi_j(k)p_{i,j}(k) + (1 - \varphi_j(k))G_{bj}(k), (\varphi_j(t) \in U(0,1)). \tag{10}$$

The IQPSO algorithm assumes that there is a one-dimensional potential well at the local attraction of each dimension and that every particle in the group has quantum behavior. Luo et al. [32] demonstrated that the interaction of individual and public search of each particle is pulled towards its local attractor $q_i = \{q_{i,1}, q_{i,2}, \ldots, q_{i,N}\}$ to ensure convergence. The probability density function of each particle flight position can be derived from the Schrodinger equation. The next step of the IQPSO iteration is described

$$X_{i,j}(k+1) = q_{i,j}(k) \pm (h_j(k)/2)\ln(1/\beta), (\beta = rand(0,1)) \tag{11}$$

where $h_j(k)$ is delta potential well characteristic length, the average position of all the particles is expressed as $r_j(k)$, defined as $h_j(k) = 2\alpha|r_j(k) - X_{i,j}(k)|$, $r_j(k) = \frac{1}{o}\sum_{i=1}^{o} p_{i,j}(k)$,

where $\alpha$ is a parameter named the 'contraction-expansion' coefficient [31]. It is the only parameter need to be set artificially, which is concerned with the convergence performance of the algorithm.

The improved methods for $\alpha$ include the fixed value and linear variation methods. In this paper, we adaptively adjust $\alpha$ according to the iterative steps within the upper and lower bounds to ensure that the algorithm obtains better global and local search ability in the whole iteration. When $\alpha = 1.78$, IQPSO can achieve better global convergence; when $\alpha < 0.5$, IQPSO is difficult to converge. So $\alpha \in [0.5, 1.8]$ is choosen, the adaptive formula of $\alpha$ with the number of iteration steps is

$$\alpha_{k+1} = \begin{cases} 1.8 - \omega(\alpha_k - 0.5)\cos(k\pi/K), & if \quad \alpha_k > 1.2 \\ 0.5 + \omega(\alpha_k - 0.5)\cos(k\pi/K), & if \quad \alpha_k \leq 1.2 \end{cases} \qquad (12)$$

where $K$ is the maximum iterative step size, $\omega$ is the adaptive adjustment coefficient, which is formulated as

$$\omega = \begin{cases} 1, & if \quad \alpha_{k+1} > 1.2 \\ 0.88, & if \quad \alpha_{k+1} \leq 1.2 \end{cases} \qquad (13)$$

In addition, different from the traditional way of closest integer and binary conversion, we use the swap operator in [31] to optimize the mobile charging sequence in this paper.

The swap operator ($SO$) is defined as $SO(m_1, m_2)$, $m_1, m_2 \in [1, N]$, which can swap the node $s_{m_1}$ and node $s_{m_2}$ in $\Phi$. So, $\Phi' = \Phi \oplus SO(m_1, m_2)$, $\Phi'$ is a new mobile charging sequence, $\oplus$ is the function of changing $\Phi$ to $\Phi'$ via $SO$. Multiple $SO$ compose a subset of swap operators ($SSO$), where $SSO = (SO_1, SO_2, \ldots, SO_w)$, $w$ is the number of $SOs$.

If $SSO1$ and $SSO2$ act successively on the $\Phi$ to get another $\Phi''$, we can combine the SSO1 and SSO2 to get a new one, which is $SSO''=SSO1 \otimes SSO2$, where $\otimes$ means merging two SSO. At this time, $\Phi'' = \Phi \oplus SSO''$. The subset with the least swap operators is named the subset of the basic swap operator ($SBSO$). Based on the above, the changes of $SBSO$ can be extended. We assume $SBSO = \Phi \odot \Phi'$, the $SBSO$ is expected to find, which can change charging sequence from $\Phi$ to $\Phi'$. $\odot$ means the function of finding $SBSO$ from $\Phi$ to $\Phi'$. Applying the $SO$ algorithm to IQPSO, (10) is updated as

$$p_{i,j}(k) = \varphi_j(k)p_{i,j}(k) + (1 - \varphi_j(k)) \oplus G_{g_j}(k) \pm \alpha|\frac{1}{o}\sum_{i=1}^{o} r_j(k) \odot X_{i,j}(k)| \ln[1/\beta]. \qquad (14)$$

For the MSSQ problem, the execution process of the IQPSO algorithm is as follows: (1) initialize parameters such as particle swarm $o$, the maximum number of iterations $K$ and minimum error $e_{min}$ of algorithm termination; (2) calculate the $TSCNS$ of each particle and set it as the personal best value to find the current best position $p_i(k)$ of each particle; (3) according to $p_i(k)$ get the global best position $G_b$; (4) update the current $p_i(k)$ and $G_b$ by comparing with the historical information; (5) update particles' positions; (6) repeat 2–5 until $e_{min}$ is met or $k = K$; (7) output global best $TSCNS$ and $G_b$. The pseudocode is shown in Algorithm 2.

---

**Algorithm 2** IQPSO algorithm for the MSSQ.

---

**Require:** $\Phi(0)$, $D$, $E_i$, $V_{cs}$, $e_m$, $v_m$, $v_c$, $o$, $K$, $e_{min}$
**Ensure:** optimal $\Phi^*$ and $TSCNS(\Phi^*)$

 1: **for** $k = 1 : K$ **do**
 2:    **for** $i = 1 : o$ **do**
 3:        Initialize positions randomly $X_i(k)$
 4:        Update the $p_i(k)$ according to the $TSCNS(p_i(k))$ and $TSCNS(X_i(k))$ via (8)
 5:    **end for**
 6:    Update the global best position $G_b(k)$ via (9)
 7:    To get the evolution equation of particles $p_i(k)$ from (14)
 8:    From (12) to (13), adaptively choosing $\alpha(k)$ according to the iteration steps
 9:    Calculate the new position of particles $\tilde{X}(k+1)$ according to (11)
10:    **if** $e_{min} <= e_{min}$ **then**
11:        Break
12:    **end if**
13: **end for**
14: $\Phi^* = G_b(K)$

---

*5.2. Iqpso Convergence Analysis*

The principle of the IQPSO algorithm is the same as that of QPSO; therefore, the proof of the convergence of QPSO is given in [31], and we will not repeat it here. In this part, we analyze the computational complexity of IQPSO.

**Theorem 2.** *The computational complexity of IQPSO is $O(N^2oK)$.*

**Proof.** The calculation of $TSCNS$ within a charging cycle needs to be for the charge of $N$ nodes. The time complexity of the IQPSO is $O(N^2)$. The IQPSO takes $O(N^2o)$ times to traverse the largest $TSCNS$ of all particles in the inner, while the loop needs to be executed $o$ times. Since the outer loop requires at most $K$ iterations, the computational complexity of IQPSO is $O(N^2oK))$.  □

In addition, we used Matlab2021b simulation software to verify the IQPSO algorithm convergence. The simulation settings in this paper are based on [33], and the specific parameters are shown in Table 1.

**Table 1.** Simulation settings.

| Parameters | Value |
|:---:|:---:|
| $L$ | 100 |
| $N$ | 80 |
| $v_c$ | 40 |
| $v_{cs}^i$ | 0.2–1.8 |
| $e_m$ | 144 |
| $v_m$ | 5 |
| $v_l$ | 1 |
| $E_m$ | 23,040 |
| $R_m$ | 6 |
| $e_{in}^i$ | 43.2–51.84 |
| $K$ of the QPSO and IQPSO | 40 |
| $o$ of the QPSO and IQPSO | 100 |

As shown in Figure 2a, the x-axis is the number of iteration steps and the y-axis represents the performance index. The IQPSO is proved to be convergent, converging to the maximum $TSCNS$ value of 58.69 in about 12 steps. The maximum $TSCNS$ value of the QPSO is 58.63 in about 19 steps; therefore, the convergence performance of IQPSO is better than that of QPSO. Similarly, as shown in Figure 2b, IQPSO before iterating 12 steps

has a large error fluctuation to increase the iteration step size and thus converges to the maximum *TSCNS* faster.

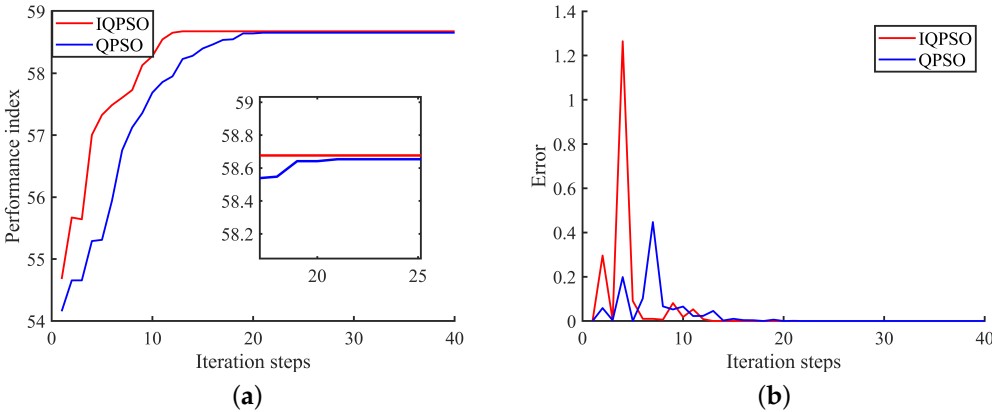

**Figure 2.** Iterative process of the IQPSO. (**a**) Performance index. (**b**) Error.

In order to make our conclusion more convincing, we have performed 20 comparison simulations, and the results are shown in the following Tables 2 and 3. The position distribution and initial residual energy of 80 nodes in 20 simulations are randomly set; therefore, the results are different each time. Finally, we averaged the results of 20 comparison simulations to obtain the performance indexes (60.25 and 59.16) and the convergence steps (12 and 17.6) of the IQPSO and QPSO algorithms, respectively. It is clear that the IQPSO is better than QPSO in convergence performance, especially for the convergence speed.

**Table 2.** Comparison of convergence speed.

|  | 1 | 2 | 3 | 4 | 5 | 6 | 7 | 8 | 9 | 10 |
|---|---|---|---|---|---|---|---|---|---|---|
| IQPSO | 9 | 12 | 15 | 8 | 11 | 14 | 13 | 11 | 10 | 15 |
| QPSO | 15 | 19 | 17 | 15 | 17 | 19 | 19 | 15 | 13 | 19 |
|  | **11** | **12** | **13** | **14** | **15** | **16** | **17** | **18** | **19** | **20** |
| IQPSO | 11 | 18 | 15 | 12 | 8 | 12 | 16 | 8 | 10 | 12 |
| QPSO | 21 | 22 | 16 | 17 | 18 | 15 | 19 | 16 | 19 | 21 |

**Table 3.** Comparison of *TSCNS*.

|  | 1 | 2 | 3 | 4 | 5 | 6 | 7 | 8 | 9 | 10 |
|---|---|---|---|---|---|---|---|---|---|---|
| IQPSO | 55.30 | 58.08 | 65.64 | 63.22 | 62.58 | 58.52 | 58.93 | 62.53 | 61.15 | 63.68 |
| QPSO | 54.35 | 57.22 | 63.13 | 62.72 | 61.88 | 58.42 | 58.89 | 61.86 | 59.26 | 63.01 |
|  | **11** | **12** | **13** | **14** | **15** | **16** | **17** | **18** | **19** | **20** |
| IQPSO | 59.02 | 60.37 | 61.48 | 55.79 | 61.98 | 59.82 | 59.29 | 58.55 | 59.99 | 59.05 |
| QPSO | 58.61 | 59.28 | 60.88 | 55.70 | 60.11 | 57.02 | 58.90 | 57.89 | 58.34 | 56.64 |

## 6. Comparative Performance Analysis

In this section, to evaluate the performance of the IQPSO algorithm, we will conduct three sets of fifty simulation tests. We analyzed the impact of the IQPSO algorithm performance on the network QSC; the average network coverage rate and the number of off-working nodes under different network scales; the MC charging powers; and the MC moving speeds, respectively. The simulation results are taken as the average of 50 tests to make the simulation results more persuasive.

### 6.1. Experimental Details

In this paper, the simulation parameters are shown in Table 1, some details are supplemented. To increase the proposed algorithm's adaptability to different WRSNs, we

assumed that 80 nodes are randomly and uniformly distributed in a fixed $100 \text{ m} \times 100 \text{ m}$ 2D monitoring plane, and the initial residual energy of each node $e_i$ is randomly generated between $30\% e_m$–$60\% e_m$. The energy consumption of each node, $v_{cs}$, is randomly generated between (0.2–1.8) J/s. In addition, the MC moving speed is 5 m/s, the charging power is 40 J/s, and the energy consumption rate of the moving unit distance is 1 J/m. We have conducted a 50-h simulation test on four NVIDIA GeForce GTX 2080 servers for the verification of the proposed algorithm and different comparison simulations.

### 6.2. Baseline Algorithms

In this part, the proposed algorithm's performance will be compared with two typical heuristic algorithms: the QPSO and Greedy algorithms [34]. The survival rate of the network nodes and the selection of nodes with larger AoIS will affect network QSC; therefore, regarding the Greedy algorithm, we designed Greedy-E and Greedy-A algorithms.

(1) The QPSO algorithm is an iterative optimization algorithm. The particle represents a possible charging sequence. Particles have quantum behavior, and their action is affected by potential quantum gravity. The QPSO algorithm updates the charging sequence by tracking the individual and global optimal performance indicators found by itself and the population.

(2) The MC executes a greedy charging strategy in the Greedy algorithm according to a specific network index. In the Greedy-E algorithm, MC always charges nodes according to the order of the remaining energy of the node.

(3) Different from the Greedy-E, the MC of the Greedy-A algorithm sorts the AoISs of the nodes to be charged in real-time from large to small and continuously selects the node with the largest real-time AoIS to charge.

### 6.3. Comparison of Network Scales

On the premise that the MC charging power remains unchanged, increasing the number of nodes will directly affect the charging decision. In this part, we will analyze the changes in the network QSC, the average coverage rate, and the number of off-working nodes for $N = \{30, 50, 70, 90, 110, 130\}$ within the fixed monitoring range (as shown in Figure 3).

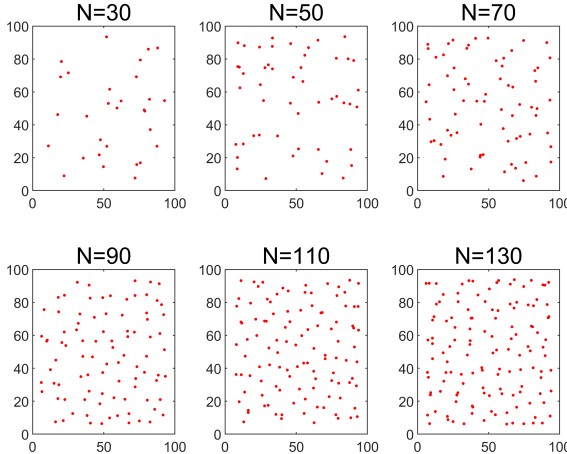

**Figure 3.** Different network scales.

As shown in Figure 4a, the network QSC gradually enhances with the increase in $N$. When $N \leq 50$, the suboptimal mobile charging sequences obtained by the four algorithms are almost the same. The MC charging power approximately meets the overall network demand. At $N = 30$, both the IQPSO and QPSO algorithms can achieve full network coverage in the charging process (Figure 4b). The WRSN works continuously, so the $TSCNS$ of IQPSO and QPSO can reach 30. As $N$ continues to increase, the charging power of the MC does not meet the overall consumption of the network, increasing the number of

off-working nodes, as shown in Figure 4c. At $N = 70$, the number of off-working nodes of the IQPSO and QPSO algorithms are highly similar, but it has obvious advantages in optimizing the network QSC and average coverage rate. $\delta_1 > \delta_2$, thus the MC pays more attention to the network sensing coverage than the number of off-working nodes and the IQPSO algorithm will charge the nodes with a larger AoIS and ignore the cost of the large number of off-working nodes. Therefore, the mobile charging sequence optimization performance in IQPSO is superior to the other heuristic algorithms in different scale networks. The more extensive the network, the more obvious the advantages.

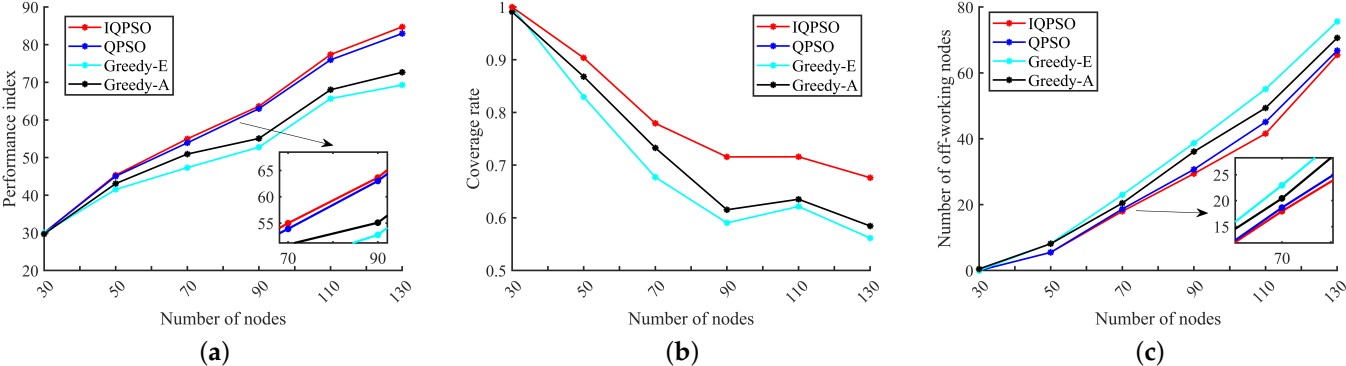

**Figure 4.** Effect of different network scales. (**a**) Performance index. (**b**) Coverage rate. (**c**) Number of off-working nodes.

### 6.4. Comparison of Charging Powers

The charging efficiency of MC is mainly affected by its charging power. If a single MC is used to ensure the sustainable work of all nodes, it has very high requirements. In this part, we will analyze the changes in the network QSC, the average coverage rate, and the number of off-working nodes for $v_c = \{10, 20, 40, 60, 80, 100\}$.

It can be intuitively seen from Figure 5a that the network QSC enhances with the increase in $v_c$. As shown in Figure 5b, when $v_c \geq 100$, in the IQPSO and QPSO algorithms, the MC can realize 'full-charge instantly', and the network will consistently achieve perfect sensing coverage. When $v_c \leq 40$, the charging power is severely limited, and the nodes with insufficient remaining energy or off-working nodes cannot be charged in time. Compared with the other three algorithms shown in Figure 5c, with the decrease in $v_c$, the number of off-working nodes increases linearly. The minimum number is 20 and the maximum is 64. At this time, the IQPSO algorithm can find the approximate optimal mobile charging sequence with obvious advantages. When the $v_c$ increases to a certain extent, its charging strategy gradually approaches Greedy-A.

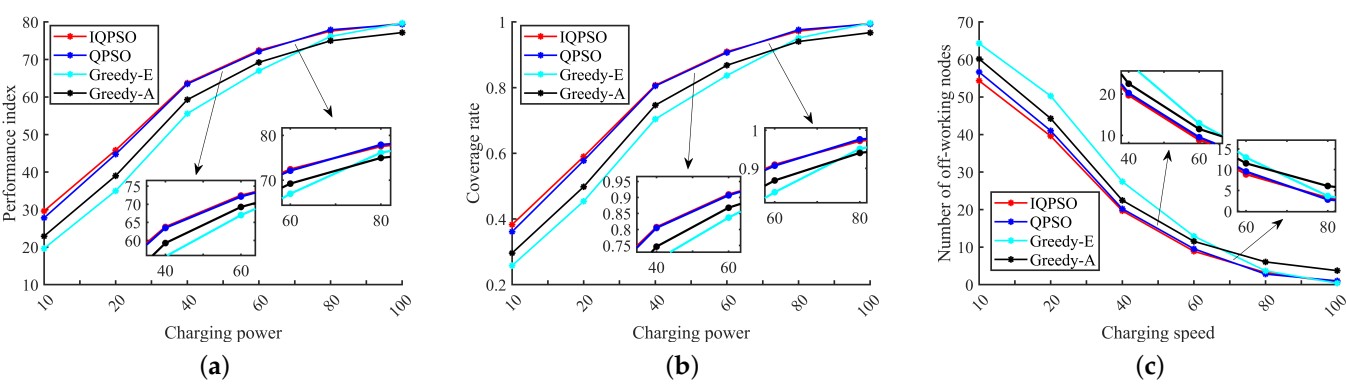

**Figure 5.** Effect of different MC charging powers. (**a**) Performance index. (**b**) Coverage rate. (**c**) Number of off-working nodes.

*6.5. Comparison of MC Moving Speeds*

The moving speed also has an important influence on the charging efficiency. The faster the $v_m$, the shorter the waiting time of nodes. In this part, we will analyze the changes in network QSC, the average coverage rate, and the number of off-working nodes for $v_m = \{1, 3, 5, 7, 9, 11\}$.

It can be seen from Figure 6a that compared with the other three algorithms, the IQPSO algorithm can still learn the suboptimal mobile charging sequence to ensure the optimal network QSC. With the increase in $v_m$, although the fluctuation in network QSC obtained by the four algorithms is relatively stable, the average network coverage rate and the number of off-working nodes both increase and decrease slowly. As shown in Figure 6b,c, in the IQPSO algorithm, the average network coverage and the number of off-working nodes are superior to the other three algorithms. In this paper, the network energy consumed during MC moving accounts for less than $e_m$. As $v_m$ increases from one to eleven, $t_l$ shortens and the energy consumed by the nodes decreases. The average network coverage rate and the number of off-working nodes increase and decrease gradually, eventually becoming stable. If $v_m$ continuously increases, $t_l \to 0$, and the energy consumed by the nodes can be ignored. Therefore, in the MSSQ problem, $v_m$ has little impact on network QSC.

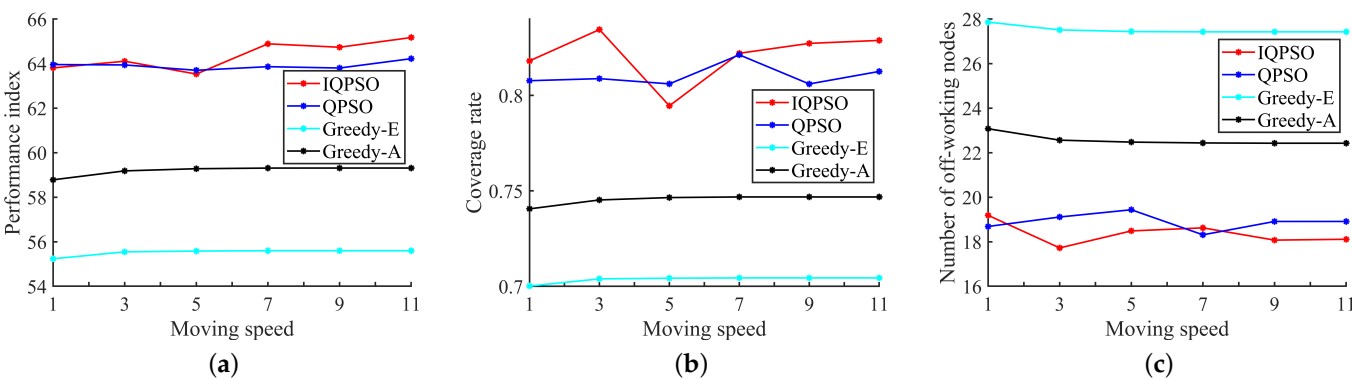

**Figure 6.** Effect of different MC moving speeds. (**a**) Performance index. (**b**) Coverage rate. (**c**) Number of off-working nodes.

## 7. Discussion

The results of this study demonstrate the effectiveness of the proposed IQPSO algorithm in improving QSC in WRSNs, especially for large-scale networks. This is an important study, as previous research on mobile charging sequence optimization in WRSNs has primarily focused on the charging efficiency as a performance index, ignoring the importance of network QSC. Moreover, previous research on the network QSC optimization has been studied in energy-harvesting WSNs and has yet to be considered in WRSNs with mobile charging.

The limitations of this study include the following: (1) IQPSO is an off-line optimization algorithm that can only be implemented in WRSNs where the state is known and highly predictable. It cannot effectively find the suboptimal mobile charging sequence for WRSNs with dynamic changes in the actual state. (2) This paper does not consider other factors affecting network QSC, such as node mobility or MC carrying capacity. Future work should address these limitations by designing an online optimization algorithm based on the network dynamic characteristics and considering the impact of other factors on network performance.

Compared with other studies, the uniqueness of this study is that the impact of the mobile charging sequence on network QSC is considered. The results found that the IQPSO algorithm is superior to other common algorithms regarding its global search ability and convergence speed.

The results of this study have practical implications for the design of MC mobile charging in WRSNs. Optimizing the mobile charging sequence can ensure that all nodes

are charged efficiently and that the network maintains a high level of network QSC in practical applications.

## 8. Conclusions

Considering the impact of mobile charging sequences on network QSC, this paper studies the novel MSSQ problem and proposes an IQPSO algorithm. In addition, we propose a new performance index for the network QSC that considers both sensing coverage and the survival rate of nodes. The proposed IQPSO algorithm adaptively adjusts the contraction expansion coefficient to obtain better global search capability and improve the convergence speed of the algorithm effectively. Through extensive simulation experiments, the IQPSO algorithm can achieve excellent performance improvements in network sensing coverage, especially in large-scale WRSNs. The state in real WRSN applications changes dynamically according to the real-time demand; therefore, designing online mobile charging scheduling algorithms for the optimal network QSC is an important research trend.

**Author Contributions:** Conceptualization, J.L., T.X. and W.X.; methodology, J.L.; software, J.L. and C.J.; validation, J.L. and C.J.; formal analysis, J.L.; investigation, J.L. and J.W.; data curation, J.L. and C.J.; writing—original draft preparation, J.L.; writing—review and editing, T.X. and W.X.; supervision, T.X. and W.X.; project administration, W.X. All authors have read and agreed to the published version of the manuscript.

**Funding:** This work was supported in part by the National Natural Science Foundations of China (NSFC) under grant 62173032, the Foshan Science and Technology Innovation Special Project under grant BK22BF005, and the Regional Joint Fund of the Guangdong Basic and Applied Basic Research Fund under grant 2022A1515140109.

**Institutional Review Board Statement:** Not applicable.

**Informed Consent Statement:** Not applicable.

**Data Availability Statement:** Not applicable.

**Conflicts of Interest:** The authors declare no conflict of interest.

## Abbreviations

The following abbreviations are used in this manuscript:

| | |
|---|---|
| WRSNs | Wireless rechargeable sensor networks |
| WSNs | Wireless sensor networks |
| MC | Mobile charger |
| QSC | Quality of sensing coverage |
| QoI | Quality of information |
| MSSQ | Mobile charging sequence scheduling for network QSC |
| ETSP | Extended TSP |
| QPSO | Quantum-behaved particle swarm optimization |
| IQPSO | Improved quantum-behaved particle swarm optimization |
| SN | Sink node |
| CS | Charging station |
| CTS | Charing time step |
| AoCS | Area of cooperation sensing |
| AoIS | Area of independent sensing |
| TSR | Total sensing range |
| SCNS | Sensing coverage and node survival rate |
| TSCNS | Total SCNS |
| MSSQ-P | Particular case of MSSQ |

## Nomenclature

| | |
|---|---|
| $R_m$ | Radius of nodes coverage (m) |
| $e_r^i$ | Residual energy of $s_i$ (J) |
| $v_{cs}^i$ | Energy consumption of $s_i$ (J/s) |
| $t_l$ | Time of MC moving (s) |
| $v_m$ | Speed of MC moving (m/s) |
| $d$ | Distance between nodes (m) |
| $a_c^i$ | AoCS of $s_i$ ($m^2$) |
| $a_v^i$ | Average AoCS of $s_i$ ($m^2$) |
| $a_{TI}$ | Total AoIS of all nodes ($m^2$) |
| $a_c^{i,j}$ | AoCS of $s_i$ and $s_j$ ($m^2$) |
| $l_s$ | Number of off-working nodes |
| $Q_i$ | $a_c^{i,j}$ set matrix of $s_i$ |
| $l_{Q_i}$ | Dimension of $Q_i$ |
| $N$ | Number of nodes in WRSNs |
| $E_m$ | MC carrying capacity (kJ) |
| $e_{in}^i$ | Initial energy of $s_i$ (kJ) |
| $t_w$ | Time of MC working (s) |
| $t_c$ | Time of MC charging (s) |
| $v_c$ | Charging power of MC (J/s) |
| $e_m$ | Capacity of nodes energy (kJ) |
| $a_{TC}$ | Total AoCS of all nodes ($m^2$) |
| $a_m$ | Maximum sensing coverage area of each node ($m^2$) |
| $a_c^{i,j,k}$ | AoCS of $s_i$, $s_j$ and $s_k$ ($m^2$) |
| $\Phi$ | MC mobile charging sequence |
| $q_i$ | $a_c^{i,j,k}$ set matrix of $s_i$ |
| $l_{q_i}$ | Dimension of $q_i$ |
| $v_l$ | MC's energy consumption per unit moving distance (J/m) |
| $a_{TSA}$ | Total sensing area ($m^2$) |

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
