# Peer review of "Mobile Charging Sequence Scheduling for Optimal Sensing Coverage in Wireless Rechargeable Sensor Networks"

_applsci, doi:10.3390/app13052840_

Round 1

Reviewer 1 Report

1. Mention your clear contribution.

2. Lack of literature review , existing work limitations did not well mentioned.

3. Algorithm did not well explain.

4. Figure 1 need more explanation.

5.experimental setting did not well explained

Author Response

Responses to Reviewer 1 Comments

Point 1:  Mention your clear contribution.

Response 1: Thank you for your suggestion. We added our contributions at the end of Section 1 in the revised manuscript, including the following three points:

  1. In this paper, we consider the impact of MC mobile charging sequence on network quality of sensing coverage (QSC) in wireless rechargeable sensor networks (WRSNs), and a novel mobile charging sequence scheduling for network QSC (MSSQ) problem is studied.
  2. The MSSQ problem is formulated as an extended traveling salesman problem (ETSP), based on a new performance index for network QSC, named total sensing coverage and node survival rate (TSCNS), which considers both the sensing coverage and node survival rate.
  3. An effective improved quantum-behaved particle swarm optimization (IQPSO) algorithm is proposed for MSSQ to obtain the suboptimal mobile charging sequence faster, and avoid falling into local optima.

Point 2:  Lack of literature review, existing work limitations did not well mention.

Response 2: According to your comments, we modified the literature review to mention the limitations of the existing work in Section 2 in the revised manuscript, including:

  1. primarily focused on charging efficiency as the performance index and ignored the important network coverage.
  2. network QSC optimization has been studied in energy-harvesting WSNs and has not been addressed in mobile charging enabled WRSNs .
  3. the computational overhead will increase significantly with the scalability of the network, and the existing algorithms will be slow if applied to the MSSQ problem.

In addition, several references [22-25] on the research of multi-node and multi-MC charging under the offline charging method were added in Section 2 in the revised manuscript.  

Point 3:  Algorithm did not well explain.

Response 3: As suggested, we explained the algorithm in details in Section 5.1 in the revised manuscript.

Point 4:  Figure 1 need more explanation.

Response 4: Thank you for the comments, in Section 3 of the revised manuscript, we gave more explanation for Figure 1, including the functions of the sink node (SN) and the charging station (CS), as well as the mobile charging sequential scheduling process.

Point 5:  Experimental setting did not well explain.

Response: As suggested, we added the simulation settings in Section 6.1 in the revised manuscript. In details, 80 nodes are randomly and uniformly distributed in the fixed 100m*100m 2-D monitoring plane, the initial residual energy of each node ei, is randomly generated between 30%em-60%em. The energy consumption of each node vcs is randomly generated between (0.2-1.8) J/s. In addition, MC moving speed is set at 5m/s, charging power is set at 40J/s, and the energy consumption rate of moving unit distance is set at 1J/m.

Reviewer 2 Report

The English writing of the paper must be considerably improved. There are many sentences that have not correct grammar and/or where words are missing, so some sentences are not fully understandable.

As the manuscript contains (too) many abbreviations, the authors should provide a list of abbreviations.

The conclusions should be more revealing on the one hand and the authors should give suggestions for general improvements by potential future works and they should not describe what they intend to do in future themselves, this is not of interest for the reader.

The references are not given in a consistent style and there are some mistakes. The authors should check and correct the references carefully and provide a consistent style that is equal to the style of Appl. Sci.

Author Response

Responses to Reviewer 2 Comments

Point 1:  The English writing of the paper must be considerably improved. There are many sentences that have not correct grammar and/or where words are missing, so some sentences are not fully understandable.

Response 1: Thanks for the comments, as suggested, the English writing of the paper have been improved significantly in the revised manuscript, including the grammar errors, missing words and the sentences that are not fully understandable.

Point 2:  As the manuscript contains (too) many abbreviations, the authors should provide a list of abbreviations.

Response 2: As suggested, a list of abbreviations is provided at end of the revised manuscript. 

Point 3:  The conclusions should be more revealing on the one hand and the authors should give suggestions for general improvements by potential future works and they should not describe what they intend to do in future themselves, this is not of interest for the reader.

Response 3: Thank you for pointing out this problem, we have revised the manuscript to make conclusions more revealing and give suggestions for general improvements by potential future works. Specifically, as in the real wireless rechargeable sensor network (WRSN) applications, the real-time demand and the environment of the network may change dynamically, how to design an online mobile charging sequence scheduling algorithm for the optimal network quality of sensing coverage (QSC) for dynamic WRSNs is challenging and can be addressed as the future work.

Point 4:  The references are not given in a consistent style and there are some mistakes. The authors should check and correct the references carefully and provide a consistent style that is equal to the style of Appl. Sci.

Response 4: Thanks for the suggestion. In the revised manuscript, the references are given in a consistent style and the mistakes are corrected, according to the style of Appl. Sci.

Reviewer 3 Report

The manuscript entitled "Mobile Charging Sequence Scheduling for Optimal Sensing Coverage in Wireless Rechargeable Sensor Networks," requires more work to substantiate the conclusions of the manuscript. Below are my further comments:

1. Provide a 'graphical abstract' and 'highlight the contributions' of
the manuscript
2. The manuscript requires further English proof from the native English speaker
3. Improve the Literature review of the manuscript, it has lacking of relevant
recent literature in the reference. Further, the direction of the problem
is not clear enough
4. The contributions and innovations from this manuscript isn't clear.
What are the innovations of your research as compared to other papers
published recently.  
5. Provide convincing results by presenting further graphical and table of comparisons, table of differences in terms of convergence speed, implementation issues, complexities, drawbacks & dependencies, error budget analysis, among others. Compare your results with the recently published papers in the literature.
6. Provide a 'proof of stability' on section 4 of the manuscript
7. A formal "statistical tests" and comparison of 'statistical results'
should be addressed in the manuscript
8. Summarize all the variables that you used in the manuscript and
cite the sources
9. Improve the references of the manuscript, use the recently published articles and Host journal as well

Author Response

Responses to Reviewer 3 Comments

Point 1: Provide a 'graphical abstract' and 'highlight the contributions’ of the manuscript.

Response 1: Thanks for your comments. As suggested, the following 'graphical abstract' and the 'highlight the contributions’ of the manuscript has been provided in the revised manuscript.

Fig 1 Graphical abstract

         Specifically, the following contributions are highlighted:

  1. In this paper, we consider the impact of MC mobile charging sequence on network quality of sensing coverage (QSC) in wireless rechargeable sensor networks (WRSNs), and a novel mobile charging sequence scheduling for network QSC (MSSQ) problem is studied.
  2. The MSSQ problem is formulated as an extended traveling salesman problem (ETSP), based on a new performance index for network QSC, named total sensing coverage and node survival rate (TSCNS), which considers the sensing coverage and node survival rate.
  3. An effective improved quantum-behaved particle swarm optimization (IQPSO) algorithm is proposed for MSSQ to obtain the suboptimal mobile charging sequence faster,and avoid falling into local optima.

Point 2: The manuscript requires further English proof from the native English speaker.

Response 2: Thanks for the comments, as suggested, the English writing of the paper have been improved significantly in the revised manuscript by the help of a native English speaker, including the grammar errors, missing words, and some sentences that not fully understandable.

Point 3: Improve the Literature review of the manuscript, it has lacking relevant recent literature in the reference. Further, the direction of the problem is not clear enough.

Response 3: As suggested, we have improved the literature review in the revised manuscript and added relevant recent publications [22-25] in the references. The specific modifications are given in Section 2. In addition, we moved the related work on network quality of information (QoI) from Section 1 to Section 2 to make the literature review more comprehensive.

The research direction of this manuscript, i.e., mobile charging sequence scheduling for optimal network sensing coverage, has been specified in Section 1 and at the end of Section 2 in the revised manuscript.

Point 4: The contributions and innovations from this manuscript isn't clear. What are the innovations of your research as compared to other papers published recently.

Response 4: As suggested, we have summarized our main contributions and innovations at the end of Section 1, including the following three points:

  1. In this paper, we consider the impact of MC mobile charging sequence on network quality of sensing coverage (QSC) in wireless rechargeable sensor networks (WRSNs), and a novel mobile charging sequence scheduling for network QSC (MSSQ) problem is studied.
  2. The MSSQ problem is formulated as an extended traveling salesman problem (ETSP), based on a new performance index for network QSC, named total sensing coverage and node survival rate (TSCNS), which considers the sensing coverage and node survival rate.
  3. An effective improved quantum-behaved particle swarm optimization (IQPSO) algorithm is proposed for MSSQ to obtain the suboptimal mobile charging sequence faster,and avoid falling into local optima.

Point 5: Provide convincing results by presenting further graphical and table of comparisons, table of differences in terms of convergence speed, implementation issues, complexities, drawbacks & dependencies, error budget analysis, among others. Compare your results with the recently published papers in the literature.

Response 5: As suggested, to provide convincing results, we have presented further graphical and table of comparisons, table of differences in terms of convergence speed, implementation issues, complexities, drawbacks & dependencies, and error budget analysis, and compared our results with the recently published papers. Related detailed revisions include:

we added two tables for comparing the convergence performance of IQPSO and QPSO in Section 5.2, one for the maximum convergence index in Table 2, and the other for the convergence speed (expressed in convergence steps) in Table 3.

         In addition, we redrew the iterative convergence graph for the performance index, added the convergence graph for the iterative error, and gave the proof of the NP-completeness of the problem, in Section 5.2 in the revised manuscript.

          In subsections 6.3-6.5 in the revised manuscript, we compared the performance of the proposed IQPSO with the baseline QPSO and Greedy algorithms. The results show that IQPSO is superior to the baseline algorithms with regard to the network QSC optimization performance.

Point 6:  Provide a 'proof of stability' on section 4 of the manuscript.

Response 6: Thank you for your suggestion. We have tried our best to address your suggestion, i.e., ‘proof of stability’ for the proposed algorithm. To our knowledge, there is no such proof in the published literature.

         As the addressed MSSQ problem is an optimization design problem, which is an extended traveling salesman’s problem (TSP). To our understanding, it does not involve the stability problem. Instead, we included the proof of the NP-completeness of the proposed problem. If there are any further suggestions, we will consider them seriously and carefully.

Point 7:  A formal "statistical tests" and comparison of 'statistical results' should be addressed in the manuscript.

Response 7: Thanks for your comments, as suggested, we added the 'statistical tests' and introduced the statistical content of the simulations in the first paragraph of Section 6, including the network QSC, average network coverage rate and the number of off-working nodes under different network scales, MC charging powers and MC moving speeds, respectively. For each set of simulation, we did 50 sets of experiments and taken the simulation results as the average of 50 tests to make it more persuasive.

The statistical results are analyzed and addressed in Sections 6.3, 6.4 and 6.5, respectively.

Point 8: Summarize all the variables that you used in the manuscript and cite the sources.

Response 8: According to your comments, we have reorganized the commonly used variables in the Nomenclature section. The specific modifications have been added to the manuscript in lines 534-563.

         We have cited the sources of the data sources of simulation settings in Section 5.2, which is published in Entropy Journal in 2022 by our research group (as reference [33] in the revised manuscript).

Point 9:  Improve the references of the manuscript, use the recently published articles and Host journal as well.

Response 9: Thank you for your comments, we have improved the references by deleting several earlier references and adding the following recent ones:

[3]. Zheng, Y.; Lu, R.; Zhang, S.; Guan, Y.; Shao, J.; Zhu, H. Toward Privacy-Preserving Healthcare Monitoring Based on Time-Series Activities Over Cloud. IEEE Internet of Things Journal 2022, 9, 1276–1288.

[4]. Macis, S.; Loi, D.; Ulgheri, A.; Pani, D.; Solinas, G.; Manna, S.L.; Cestone, V.; Guerri, D.; Raffo, L. Design and Usability Assessment of a Multi-Device SOA-Based Telecare Framework for the Elderly. IEEE Journal ofBiomedical and Health Informatics 2020, 24, 268–279.

[5]. Zhu, F.; Lv, Y.; Chen, Y.; Wang, X.; Xiong, G.; Wang, F.Y. Parallel Transportation Systems: Toward IoT-Enabled Smart Urban Traffic Control and Management. IEEE Transactions on Intelligent Transportation Systems 2020, 21, 4063–4071.

[6]. Zhang, C.; Zhou, G.; Li, H.; Cao, Y. Manufacturing Blockchain of Things for the Configuration of a Data- and Knowledge-Driven Digital Twin Manufacturing Cell. IEEE Internet of Things Journal 2020, 7, 11884–11894.

[10] Verma, G.; Sharma, V. A Novel Thermoelectric Energy Harvester for Wireless Sensor Network Application. IEEE Transactions on Industrial Electronics 2019, 66, 3530–3538.

[13] Mo, L.; Kritikakou, A.; He, S. Energy-aware multiple mobile chargers coordination for wireless rechargeable sensor networks. IEEE internet of Things journal 2019, 6, 8202–8214.

[14]. Wei, Z.; Xia, C.; Yuan, X.; Sun, R.; Lyu, Z.; Shi, L.; Ji, J. The path planning scheme for joint charging and data collection in WRSNs: A multi-objective optimization method. Journal of Network and Computer Applications 2020, 156, 102565.

[22] Liu, T.; Wu, B.; Zhang, S.; Peng, J.; Xu, W. An Effective Multi-node Charging Scheme for Wireless Rechargeable Sensor Networks. In Proceedings of the IEEE INFOCOM 2020 - IEEE Conference on Computer Communications, 2020, pp. 2026–2035.

[23]. Xu, W.; Liang, W.; Jia, X.; Kan, H.; Xu, Y.; Zhang, X. Minimizing the Maximum Charging Delay of Multiple Mobile Chargers Under the Multi-Node Energy Charging Scheme. IEEE Transactions on Mobile Computing 2021, 20, 1846–1861.

[24]. Sun, L.; Wan, L.; Liu, K.; Wang, X. Cooperative-Evolution-Based WPT Resource Allocation for Large-Scale Cognitive Industrial IoT. IEEE Transactions on Industrial Informatics 2020, 16, 5401–5411.

Host journal:

[29] Zhang, L.; Wang, J.; Wu, H.; Wu, M.; Guo, J.; Wang, S. Early Warning of the Construction Safety Risk of a Subway Station Based on the LSSVM Optimized by QPSO. APPLIED SCIENCES-BASEL 2022, 12.

[30]. Alajmi, M.S.; Almeshal, A.M. Least Squares Boosting Ensemble and Quantum-Behaved Particle Swarm Optimization for Predicting the Surface Roughness in Face Milling Process of Aluminum Material. APPLIED SCIENCES-BASEL 2021, 11.

[31]. Deng, Z.; Chen, T.; Wang, H.; Li, S.; Liu, D. Process Parameter Optimization When Preparing Ti (C,N) Ceramic Coatings Using Laser Cladding Based on a Neural Network and Quantum-Behaved Particle Swarm Optimization Algorithm. APPLIED SCIENCES-BASEL 2020, 10.

Reviewer 4 Report

Dear Authors

Thank you very much for allowing me to be part of the review of the work entitled: Mobile Charging Sequence Scheduling for Optimal Sensing Coverage in Wireless Rechargeable Sensor Networks.   

Wireless Rechargeable Sensor Networks (WRSN) have emerged as an alternative to solve the size and operation time challenges posed by traditional battery-powered systems.

A fundamental question in WRSN design is how to implement readers in a network to ensure that WISP tags can collect sufficient energy for continuous operation. They address the problem of energy provisioning. They propose a practical model of wireless charging supported by data. Two main aspects are point provisioning and route provisioning.

The paper is well structured according to the parameters of the journal.  The research topic is very successful, I would like to share with you some observations to improve the work done.  I will divide the comments into sections that are related to your work:

Introduction 

The introduction includes current references related to the research and it is relevant to include the objective pursued, as it provides clarity to those interested in the subject.

Although the objective of the work is mentioned in 4.1, It would be good to specify the main objective of the research at the end of the introduction.

It would be to specify in Figures 1, 2, the source.

Materials and Methods

The proposed methods are adequate

It would be good to revise the nomenclature of formulas 3, 4, 5 6. 

It would be good to include a section with all the nomenclature used in the work.

Discussion

It would be good to include a discussion section including other similar works that help to verify the fulfillment of the objective of the proposed research.

Conclusions

It would be good to include in the conclusion a paragraph indicating how the objective has been achieved?

Author Response

Responses to Reviewer 4 Comments

Point 1:  Introduction

The introduction includes current references related to the research and it is relevant to include the objective pursued, as it provides clarity to those interested in the subject.

Although the objective of the work is mentioned in 4.1, It would be good to specify the main objective of the research at the end of the introduction.

Response 1: As suggested, we have added the objective of this research at the end of the introduction (Section 1) in the revised manuscript, specifically, this paper studies the novel problem of mobile charging sequence scheduling for network QSC (MSSQ) to maximize network QSC by optimizing the mobile charging sequence, especially when the MC power is insufficient.

Point 2:  Materials and Methods

The proposed methods are adequate

It would be good to revise the nomenclature of formulas 3, 4, 5 6.

It would be good to include a section with all the nomenclature used in the work.

Response 2: Thank you for your valuable suggestion. A Nomenclature was added at the end of the revised manuscript, where all formulas you mentioned 3, 4, 5 and 6 have been explained clearly. We revised the nomenclature of formulas 6.

Point 3: It would be good to include a discussion section including other similar works that help to verify the fulfillment of the objective of the proposed research.

Response 3: Thank you for your suggestion. We have added the Discussions section (Section 7), where we described the main findings and the remain limitations of our study, highlighted the difference between the proposed algorithm and the existing methods, and gave the suggestions for the future work.

Point 4:  It would be good to include in the conclusion a paragraph indicating how the objective has been achieved?

Response 4: Thank you for your comments, we have added a paragraph in the conclusions in the revised manuscript to indicate how the objective has been achieved, i.e., by introducing a new performance index that considers both sensing coverage and the survival rate of nodes, and proposing IQPSO algorithm which can adaptively adjust the contraction expansion coefficient to obtain better global search capability and improve the convergence speed.

Round 2

Reviewer 1 Report

 Well addressed.

Reviewer 3 Report

Based on the response of the authors in the manuscript "Mobile Charging Sequence Scheduling for Optimal Sensing Coverage in Wireless Rechargeable Sensor Networks," the reviewer is satisfied with the answers and the revisions made in the manuscript.